# Integrating Genomic and Chromosomal Data: A Cytogenetic Study of *Transancistrus santarosensis* (Loricariidae: Hypostominae) with Characterization of a ZZ/ZW Sex Chromosome System

**DOI:** 10.3390/genes14091662

**Published:** 2023-08-22

**Authors:** Mauro Nirchio Tursellino, Marcelo de Bello Cioffi, Francisco de Menezes Cavalcante Sassi, Geize Aparecida Deon, Claudio Oliveira, Mariana Kuranaka, Jonathan Valdiviezo-Rivera, Víctor Hugo Gonzalez, Anna Rita Rossi

**Affiliations:** 1Departamento de Acuicultura, Universidad Técnica de Machala, Av. Panamericana km 5.5, Vía Pasaje, Machala 070150, El Oro, Ecuador; vgonzalez@utmachala.edu.ec; 2Departamento de Genética e Evolução, Universidade Federal de São Carlos, São Carlos 13565-090, SP, Brazil; mbcioffi@ufscar.br (M.d.B.C.); fmcsassi@estudante.ufscar.br (F.d.M.C.S.); geizeadeon@gmail.com (G.A.D.); 3Departamento de Biologia Estrutural e Funcional, Instituto de Biociências Universidade Estadual Paulista-UNESP, Botucatu 18618-689, SP, Brazil; claudio.oliveira@unesp.br (C.O.); mariana.kuranaka@unesp.br (M.K.); 4Instituto Nacional de Biodiversidad, Rumipamba No. 341 y Av. Shyris, Parque La Carolina, Quito 170135, Pichincha, Ecuador; bioictiojona@yahoo.com; 5Dipartimento di Biologia e Biotecnologie “C. Darwin”, Sapienza—Università di Roma, Via Alfonso Borelli 50, 00161 Rome, Italy; annarita.rossi@uniroma1.it

**Keywords:** Siluriformes, rDNA, CGH, telomeric repeats, COI, evolution

## Abstract

The plecos (Loricariidae) fish represent a great model for cytogenetic investigations due to their variety of karyotypes, including diploid and polyploid genomes, and different types of sex chromosomes. In this study we investigate *Transancistrus santarosensis* a rare loricariid endemic to Ecuador, integrating cytogenetic methods with specimens’ molecular identification by mtDNA, to describe the the species karyotype. We aim to verify whether sex chromosomes are cytologically identifiable and if they are associated with the accumulation of repetitive sequences present in other species of the family. The analysis of the karyotype (2n = 54 chromosomes) excludes recent centric fusion and pericentromeric inversion and suggests the presence of a ZZ/ZW sex chromosome system at an early stage of differentiation: the W chromosome is degenerated but is not characterized by the presence of differential sex-specific repetitive DNAs. Data indicate that although *T. santarosensis* has retained the ancestral diploid number of Loricariidae, it accumulated heterochromatin and shows non-syntenic ribosomal genes localization, chromosomal traits considered apomorphic in the family.

## 1. Introduction

The Neotropical region, which includes South and Central America, as well as the Caribbean, suffered a strong decline (−94%) in vertebrate populations, compared to 1970, and this reduction included also freshwater fish, which were globally the most harmed [1]. Despite this, tropical South America still has the greatest diversity of any region of comparable area, for numerous organisms, including many freshwater taxa [2]. Part of this richness is still unknown, with historical estimates suggesting that about 34–42% of the Neotropical freshwater fish fauna is still unrecorded [3]. Describing and recording these unknown species is crucial to prevent the risk of losing them before even acknowledging their existence. The fundamental stage in any biodiversity inventory is fish species identification. Traditionally, in fishes, it is based on morphological characteristics but has gradually been combined with molecular tools, to describe the diversity at the genomic level, i.e., as DNA barcoding [4] and cytogenetic data [5,6]. Additionally, employing an integrative approach allows for the detection of new taxa through enhanced precision and circumvents challenges associated with traditional morphological identification, including misidentification due to limited morphological variation, phenotypic plasticity, and the presence of cryptic species [7]. The cytogenetic approach allows the detection of variations in both the number and structure of chromosome complements making the identification of intra- and inter-cytotype variants possible. Indeed, fishes provide an attractive model for cytogenetic investigations due to their variety of karyotypes, including diploid and polyploid genomes, and different types of sex chromosomes systems, which provide unique insights into chromosome structure and behaviour e.g., [8,9,10]. This contributes to improving our understanding of the microevolutionary processes and also in resolving taxonomic problems, especially when morphological and meristic features make accurate species diagnosis difficult [11]. Such methods have also enabled the accumulation of information in a vast database of DNA sequences such as Genbank (https://www.ncbi.nlm.nih.gov/genbank/ (accessed on 12 June 2023)), BOLD (https://www.boldsystems.org/ (accessed on 12 June 2023)) and karyotypes [12] resulting in new tools for cataloguing biodiversity and new conservation approaches [13,14,15].

The plecos, fishes of the Loricariidae family, represent an excellent example of this sort of situation. With 115 recognized genera and more than 1000 valid Neotropical species [16], this is the most diversified family of Siluriformes (Teleostei: Ostariophysi) and the fifth most species-rich vertebrate family on Earth [17]. Most plecos are restricted to freshwater ecosystems and nearly intolerant to salinity [18], being geographically distributed in Central American drainages, south of Costa Rica, and South America, both in the Atlantic slope, north of Buenos Aires, and in the Pacific slope, north of Peru [19,20]. The taxonomy of this family was updated in the previous ten years; as a result, several genera were deemed invalid or synonymized, new genera were discovered, and species were grouped into six subfamilies [20]. The Lujans’s et al. comprehensive molecular phylogeny of the subfamily Hypostominae, which includes about half the species of the family, revealed the presence of seven distinct clades: *Peckoltia*, *Hemiancistrus*, *Acanthicus*, “*Pseudancistrus*”, *Lithoxus*, *Pseudancistrus*, and *Chaetostoma* [19]. The latter showed an evident biogeographical pattern with a monophyletic group consisting primarily of northern Andean genera (*Cordylancistrus*, *Dolichancistrus*, and *Leptoancistrus*) sister to the more widespread genus *Chaetostoma*. In addition, the presence of further genera was suggested. Indeed, two new monophyletic sister genera were later recognized, each including two species: *Andeancistrus* present in rivers draining the Amazonian slopes of the Andes Mountains in Ecuador, and *Transancistrus* which is distributed across rivers draining the Pacific slope of the Andes Mountains in Ecuador [21]. 

Only a little amount of cytogenetic information is known for loricariid catfishes; the most thorough collection of karyotypes [12] revealed information for only 7% of the recognized valid species in this family [16]. This number has grown as a result of recent research [22,23,24,25,26,27,28,29,30,31,32], albeit it is impossible to determine it precisely because the majority of the new information relates to samples that could not be reliably identified at the species level. On the whole cytogenetic data presently available include representatives of 29 genera and only one out of the 50 Loricariidae from Ecuador.

In this context, our research performs the first cytogenetic analysis of *T. santarosensis* [33] formerly *Cordylancistrus santarosensis*. This rare endemic species is only found in the Santa Rosa River (Gulf of Guayaquil drainage), close to Ecuador’s southern coast. Deforestation, agricultural and urban development, mining operations, and pollution have a significant negative impact on the ecosystem it inhabits. Although the IUCN Red List Committee [34] categorized *T. santarosensis* as Least Concern, it was noted that “total population size and population trends are unknown” and that the effect of current threats had not been thoroughly studied [35]. The cytogenetic data, based on a suite of conventional and molecular cytogenetic methods, including fluorescence in situ hybridization (FISH) with 5S and 18S ribosomal DNA (rDNA) and telomeric probes, and intra-specific complete genome hybridization (CGH), were integrated with the mitochondrial (mtDNA) sequence analysis, which allowed us to identify the specimens at the molecular level. The primary aim of this work is to describe the unknown karyotype of the species and identify species-specific cytogenetic markers that will allow us to infer which are the plesiomorphic and derived characteristics within Hypostominae. Until now, no one of the species of the *Chaetostoma* clade has been investigated, and data from species of the other clades indicate an extensive range of chromosome numbers, with 2n = 52–84. In addition, in other subfamilies, both the absence of cytologically identifiable sex chromosomes or the presence of standard (XX/XY, ZZ/ZW) or multiple sex chromosome (MSC) systems were reported [12,22,24,25,36,37,38,39,40,41,42]. Thus, we also aim to prospect the occurrence of sex chromosomes, and if they are associated with the accumulation of repetitive sequences that allows inferring the timing of their differentiation. We used comparative genomic hybridization (CGH) for this purpose, which will enable us to determine whether the non-recombination region of the sex chromosomes accumulated enough sex-limited or -enriched sequences, as observed in other fish species [43,44,45]. The molecular identification of the specimens will also allow us to compare the cytogenetic data to a known species (identified both on morphological and molecular basis) and determine if the patterns of molecular and karyotype evolution are consistent.

## 2. Materials and Methods

### 2.1. Sampling 

Twenty-five specimens of *T. santarosensis* (four males, nine females and twelve undetermined) (Appendix A) were collected with a seine net in Dos Bocas River (Cantón Pasaje), Río Palenque (Cantón Pasaje). Fish were transported to the laboratory in 32-inch sealed plastic bags containing 2 gallons of water, with the air replaced for pure oxygen, and then maintained in aquariums until processing. Ecuador’s Ministry of Environment has been permitted to collect specimens under license N° MAAAE-DBI-CM-2021-0152. The procedures were carried out following the Universidad Técnica de Machala’s Ethics Committee on Animal Experimentation (process number UTMACH-CEEA-002/2022). Voucher specimens were deposited in Ecuador’s Instituto Nacional de Biodiversidad (INABIO) fish collection and Universidad Técnica de Machala (Appendix A). Morphological Identification of specimens was performed following [20]. Four specimens of *Chateostoma bifurcum*, to be used in the molecular analysis were also collected (see Appendix A).

### 2.2. Molecular Identification of Samples and Phylogenetic Reconstruction

Genomic DNA was extracted from the muscle tissue using the Wizard Genomic DNA purification kit (Promega). The mitochondrial cytochrome oxidase I (COI) gene was partially amplified by polymerase chain reaction with the Fish F1 and Fish R1 primers [46] following [47]. Sequences were investigated using ABI PRISM 3130 Genetic Analyzer Associated software (Applied Biosystems, Foster City, CA, USA).

The alignment was performed in software Clustal X [48] using the basic local alignment search tool (BLAST, https://blast.ncbi.nlm.nih.gov/Blast/ (accessed on 12 June 2023)) to search the GenBank database for similar sequences. Also the BOLD system was explored for reference sequences. We included in analysis sequences of species of the *Chaetostoma* clade available in the GenBank database (at least two sequences for each species, when available), and *Ancistrus clementinae* was used as the outgroup (Appendix A).

Genetic distances, neighbor-joining (NJ) and maximum likelihood (ML) phylogenetic reconstruction were performed using MEGA5 [49] and 1000 bootstrap pseudoreplicates. The substitution model (HKY + G) used for ML was also calculated in MEGA5.

### 2.3. Cytogenetic Procedures

The metaphase chromosome plates were obtained from kidney cell suspensions according to the “air drying” technique [50]. Karyotypes were assembled using Giemsa-stained metaphases, and heterochromatin distribution was visualized using C-banding [51]. The nucleolar organizer regions (NORs) were detected by silver nitrate staining [52].

The modal diploid number was determined using 60 metaphases from each individual. For each sample/method, not less than 30 cells with the best metaphase chromosomes were examined. Images were edited for optimization of brightness and contrast using Photoshop (Adobe Systems, Inc. San José, CA, USA, Version 2015.0.0). To construct karyotypes, we used the arm ratio criteria to classify chromosomes as metacentric (m)), submetacentric (sm), subtelocentric (st), and acrocentric (a) [53]. To calculate the fundamental number (FN), m and sm chromosomes were considered as biarmed, while chromosomes were treated as uniarmed.

The FISH experiments were performed according to Pinkel et al. [54] with adaptations of Sassi et al. [55]. The probes of ribosomal genes 18S rDNA and 5S rDNA, and telomeric repeats (TTAGGG)n were obtained by polymerase chain reaction (PCR) with previously described primers [56,57,58]. For the comparative genomic hybridization (CGH), total DNA was extracted from male and female samples by the standard phenol-chloroform-isoamyl alcohol method [59], and the probe mix per each slide contained 500 ng of each male and female-derived DNA and 15 µg of unlabeled blocking DNA (C0t-1) which was obtained by DOP-PCR from male genome and used as a blocker to highly and moderated repeated sequences [60]. Such ratio of probe vs. C0t-1 DNA amount was chosen based on experiments performed in other fish groups [26]. Nick-Translation was used to label the probes in green with Atto488-dUTP (18S rDNA, female-derived gDNA) or in red with Atto550-dUTP (5S rDNA, telomeric sequence, and male-derived gDNA) following the manufacturer’s instructions (Jena Biosciences, Jena, Germany). Chromosomes were counterstained with 4′,6-diamidino-2-phenylindole (DAPI) mounted in Vectasield Antifade Mounting Medium (Vector Laboratories, Newark, CA, USA) and signals were visualized in an Olympus BX53 fluorescence microscope with appropriate filters).

## 3. Results

### 3.1. COI Identification and Phylogenetic Reconstruction

COI sequences (650 base pairs) obtained here for *T*. *santarosensis* (corresponding to two different haplotypes) and *C*. *bifurcum* (one haplotype) (see Appendix A) were deposited in GenBank (A.N. OR237841-43). No other COI sequence was present in the system for these species. For *T. santarosensis* BLAST function gave back the highest similarities (91.6–91.8%) with sequences belonging to *Pterygoplichthys pardalis* (although this species does not belong to the *Chaetostoma* clade, but to Hypostomini). We confirmed these results comparing on BOLD database, where no match was found for our sequences.

The phylogenetic reconstructions (Figure 1) obtained by the inclusion of the other species of the *Chaetostoma* clade [21] available in GenBank, revealed that *T*. *santarosensis* samples form a monophyletic well-supported clade both in ML and NJ trees. The genetic distances (Kimura 2-parameters distance, K2P) [61] between *T*. *santarosensis* and the other species of the *Chaetostoma* clade included in the phylogenetic analysis exceed 11%.

### 3.2. Cytogenetic Analysis

*T. santarosensis* specimens had a diploid complement of 2n = 54 chromosomes (Figure 2) and FN = 106, with differences in one chromosomal pair between individuals of the two sexes. In fact, females have a karyotype composed of 25m + 27sm+ 2a chromosomes whereas males’ karyotype is composed of 26m + 26sm + 2a chromosomes. The presence of a heteromorphic pair of chromosomes (one medium-sized and a small submetacentric) in females, suggests the presence of a ZZ/ZW sex chromosome system in the species (Figure 2a,c). This data was confirmed by C-banding showing that the putative W chromosome is nearly entirely heterochromatic. Constitutive heterochromatin blocks are also found in the peri/paracentromeric position of most chromosomes (Figure 2b,d). The longer sm chromosome pair, number 13, has a clear secondary constriction on its telomeric region present in both males and females (Figure 2e) and silver staining verified the presence of Ag-NOR in these sites (Figure 2f); the long arm of this chromosome, flanking NOR, is entirely heterochromatic (Figure 2b,d).

The rDNA probe-based double FISH experiment verified the existence of a single, major ribosomal gene cluster that corresponded to the Ag-NOR signals localized in the secondary constriction seen in the long arms of the submetacentric pair 13, in both sexes (Figure 3a). These sites are dark after DAPI staining (Figure 3a,b inset). The 5S rDNA probe, on the other hand, only detected two small ribosomal cistrons interstitially located on the short arms of a large metacentric pair (likely pair 13) in both sexes. Telomeric repeats were found at the ends of every chromosome and interstitial telomeric repeats (ITs) were not identified; (Figure 3b). The CGH experiment showed that there are no evident differences between the patterns obtained using male or female genomic DNA as probes, and thus detectable differences in repetitive DNA content in the genomes of the two sexes (Figure 4). 

## 4. Discussion

Fishes display a diverse set of chromosomal counts and genome sizes [12,62], including karyotype dynamics that can vary significantly between lineages [63]. This is linked to an astounding range of mechanisms for sex determination and differentiation [64,65,66,67,68,69] found in only about 10% of fish species investigated so far, with about half of them distributed in the Neotropical region [8]. 

This applies also to Loricariidae so that they represent a highly valuable group for evolutionary studies. Indeed, they exhibit a high degree of heterogeneity in the diploid number (2n) and chromosome formulas across subfamilies and even genera, ranging from 2n = 34 in *Ancistrus cuiabae* to 2n = 84 in *Hypostomus* sp. [37,40,70]. In addition, they show all the possible different sex chromosome systems, i.e., standard XX/XY or ZZ/ZW or even MCS systems [25,26,41,71,72,73]. The putative plesiomorphic karyotype in this family has been inferred to correspond to a diploid number 2n = 54 with one NOR-bearing pair of chromosomes, synteny of 18S and 5S rDNA sequences [22,74,75,76,77] and the absence of conspicuous heterochromatic blocks [36,74]. However, new reports of karyotypes from basal clades (Loricariinae) [27] suggested that 2n = 58 might reflect the plesiomorphic diploid number. The *Chaetostoma* clade is regarded as basal within the subfamily Hypostominae, but its position is not consistent in phylogenetic reconstructions of the Loricariidae, being closer to Loricariinae [19] or Hypoptopomatinae [17]. Thus, both the chromosome number (the first reported for the *Chaetostoma* clade) and the mapping of repetitive sequences in *T*. *santarosensis* could provide a further piece to disentangle the evolution in the subfamily and Loricariidae.

Indeed, repetitive sequences like multigene families (rDNAs), transposable elements (TEs) and satellite DNAs (satDNAs) could be used to trace chromosome rearrangements along lineages. In addition, satDNAs represent the main component of heterochromatin accumulating in the centromeric and telomeric regions of chromosome, and play a dynamic role in the evolution of B and sex chromosomes [78,79]. Here the mapping of repetitive sequences allowed us to make some inferences on the main route of genomic reorganization and chromosome rearrangements in the subfamily.

First, the diploid number (2n = 54) suggests that *T. santarosensis* kept the probable Loricariidae ancestral karyotype [74]. However, in the hypothesis that the plesiomorphic character was 2n = 58 [27,32], then the karyotype observed in *T. santarosensis* should be the result of chromosome fusions reducing its diploid number, an evidence not supported by telomeric repeats localization (see below). Also, the absence of synteny between the major and minor ribosomal genes does not match the ancestral condition reported in Loricariidae [36] although corresponds to what is observed in other Hypostominae [25]. NORs length heteromorphism can be attributed to differences in the number of cistrons and their transcriptional activity [80,81,82], due to sister chromatid unequal recombination or retrotransposition [83]. The minor ribosomal genes are localized in a single chromosome locus, a condition common to about 57% of animals, and include the 5S rDNA motif that usually is not balanced in copy number compared to 18S rDNA [84]. 

Second, constitutive heterochromatin, mainly includes satDNAs and in addition to serving as structural elements of chromosomes, keeps the integrity of the genome by inhibiting the recombination and transcription of repetitive sequences [85]. Additionally, constitutive heterochromatic regions prevent mobile element activity, isolate DNA repair in repeated sections, and guarantee proper chromosome segregation—all of which are essential for preserving genomic stability reviewed in [86]. It has been suggested that the gain of heterochromatin is a common phenomenon in chromosome evolution [87]. *T*. *santarosensis* exhibited prominent heterochromatic blocks in its chromosomes, primarily in the peri/paracentromeric position, but also in the long arm of chromosome pair 13, indicating an extensive amplification of repetitive sequences in this region, as also observed by scattered signals on CGH investigation. This feature cannot be considered an ancestral character, expected in a basal clade, as usually absence or reduced blocks of constitutive heterochromatin are observed in loricariids chromosomes [36,74]. Future studies in *T*. *aequinoctialis*, the only other species of the genus, would confirm if this characteristic is a distinctive feature of *Transancistrus*.

Third, the patterns of constitutive heterochromatin, along with variations in its distribution, could also offer valuable insights on sex chromosomes as well. Usually, the presence of sex chromosomes in teleosts is associated with their partial or full heterochromatinization [8,15,74,88,89,90,91,92,93]. The pattern observed in *T*. *santarosensis* suggests a ZZ/ZW sex chromosome system characterized by heterochromatinization of a great part of the W chromosome, as detected in other Loricariidae [94]. This type of sex chromosome system is present in other Hypostominae, such as *Ancistrus* [39,94,95], *Hemiancistrus* [96] *Hisonotus* (described in [97] as *Microlepidogaster*), *Hypostomus* [71,98,99]. The W size reduction (compared to the Z chromosome), can indicate the degeneration process of this chromosome. Indeed, the accumulation of repetitive DNA sequences on the sex-limited W or Y chromosomes is known in many fishes, mainly given the differential accumulation of rDNA and satDNAs [80,100]. This accumulation is associated with the suppression of recombination between the proto-sex pair [101], a crucial step for their differentiation in size and genetic content as observed in other fishes [102]. The presence of a prominent C-band on the W chromosome (indicating its highly heterochromatic content) does not imply that the sequences that comprise this heterochromatin are, in fact, female-specific ones. This is not uncommon in fishes, because many species, despite having morphologically heteromorphic sex chromosomes, are at an early stage of their molecular differentiation [8,73], and varying degree of the differentiation can be observed even between related species [100]. Meanwhile, the observed size difference in the Z and W chromosome of *T*. *santarosensis* aligns with the concept that chromosome differentiation may differ between male and female heterogametic systems. Additionally, it suggests that ZW systems tend to undergo faster differentiation in evolutionarily young sex chromosome systems [69].

Fourth, telomeric repeats are usually localized at the ends of chromosomes, although in vertebrates, including fishes, interstitial telomeric sequences (ITSs), can be present also along chromosome arms [103,104]. These ITSs sites are remnants of chromosome rearrangements [105,106], like chromosome fusion or pericentric inversion and can correspond to heterochromatic areas [107], and/or can overlap with NORs [108,109,110,111]. The absence of ITSs in the karyotype of *T*. *santarosensis* might be interpreted considering (a) that the abovementioned chromosome rearrangements did not represent the common mechanisms that acted in chromosome evolution in this genus, or (b) were not recent, so that after chromosomal fusion telomeric repeats (at least those localized at the centromeres) were lost or inactivated [112].

The molecular identification of the specimens confirmed the correct morphological attribution. Some differences were observed between ML and NJ in the branching of the other species within the *Chaetostoma* clade, and in the close relationships observed with *P. pardalis*, which belongs to a different clade of Hypostominae. However, here only mitochondrial sequences of a single gene were used, as a deep phylogenetic analysis was beyond the scope of this paper. Future investigations are required to clarify this aspect. For sure the karyotype of *T*. *santarosensis* is different in chromosome number, sex chromosome system and in the mapping of other repetitive sequences from that of *A. clementinae*, the unique species included in the molecular analysis for which the karyotype is known. The cytogenetic analysis of other species within the *Chaetostoma* clade will provide valuable insights into whether the cytogenetic features observed in *T*. *santarosensis* are common among other representatives of the genus and this particular clade.

## 5. Conclusions

Our results on *T. santarosensis* have unveiled both ancestral (chromosome number) and derived (heterochromatic blocks and non-syntenic ribosomal genes localization) cytogenetic characteristics in this species. Our investigation suggests that the karyotype of *T. santarosensis* has remained unaffected by recent centric fusion and pericentromeric inversion as main chromosome rearrangement mechanisms. Thus, the cytogenetic pattern is only partially congruent with the molecular pattern of evolution, indicating the *Chaetostoma* clade as basal within Hypostominae. We also identified the presence of a ZZ/ZW sex chromosome system at an early stage of differentiation. This identification raises the intriguing possibility of similar systems being present in other species of the *Chaetostoma* clade. However, the turnover of sex chromosomes in fishes does not permit us to exclude alternative pictures. Further research should extend to include not only *T. santarosesnsis* but also other Hypostominae from the Pacific slope. The goal would be to establish whether shared cytogenetic features can be linked to a discernible biogeographic pattern. Additional it is necessary to investigate the presence of TEs and other repetitive sequences within the identified sex chromosomes. 

Finally, we emphasize the significance of cytogenetics within an integrated approach for elucidating the evolutionary pathways in fishes, which is particularly significant in lesser-explored regions, such as the location of our study. Moreover, our findings contribute to enhancing the recognition of the Ecuador region as a biodiversity hotspot for fish species emphasizing the broader implications of our research.

## Figures and Tables

**Figure 1 genes-14-01662-f001:**
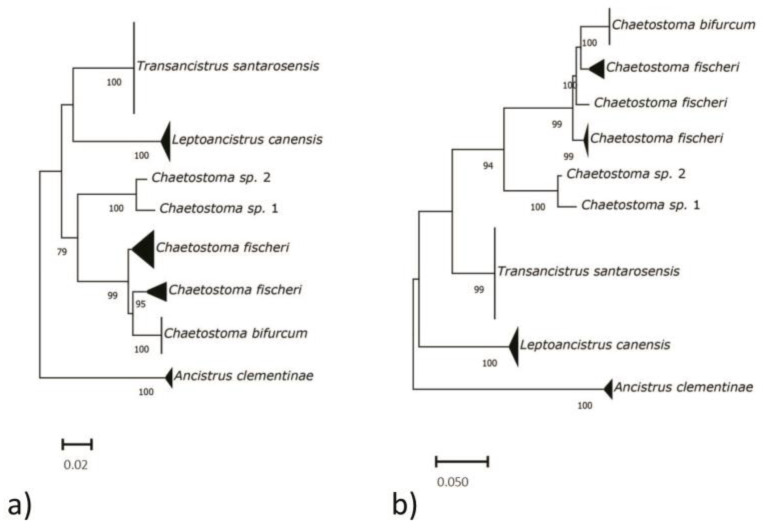
NJ (**a**) and ML (**b**) phylogenetic trees. Node support values higher than 75% are shown.

**Figure 2 genes-14-01662-f002:**
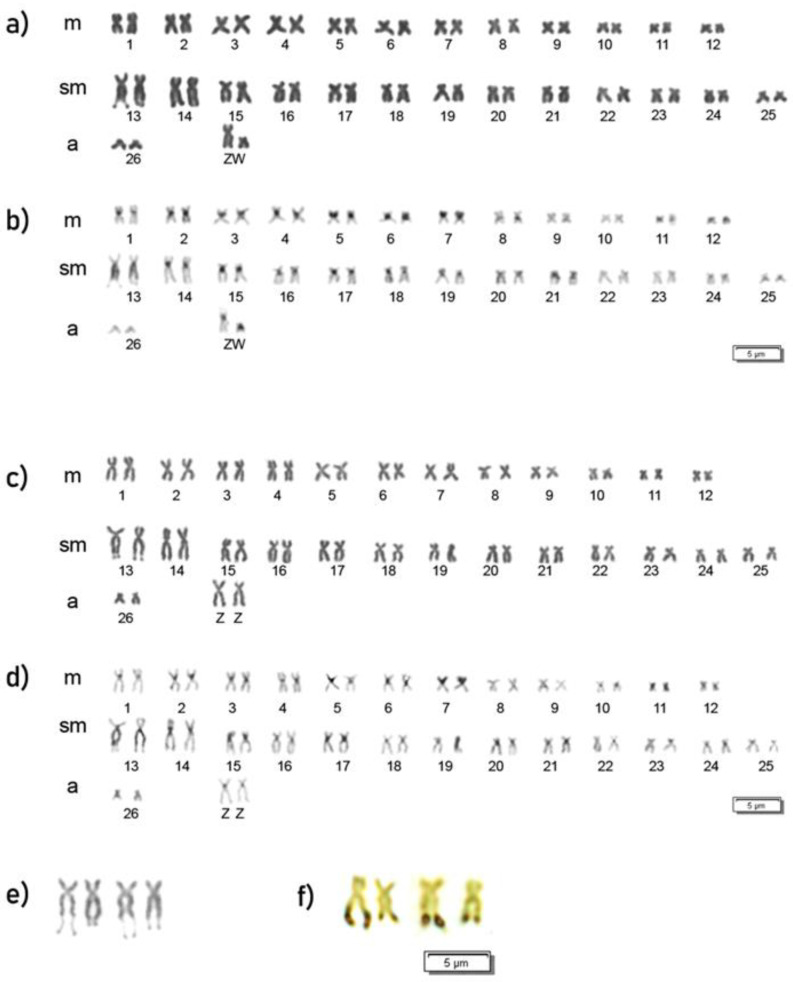
Female (**a**,**b**) and male (**c**,**d**) *T. santarosensis* karyotypes arranged after sequential Giemsa (**a**,**c**) and C-banding (**b**,**d**) staining. Enlargements of samples of chromosome pair n. 13 with evident secondary constriction in Giemsa (**e**) and after silver staining (**f**).

**Figure 3 genes-14-01662-f003:**
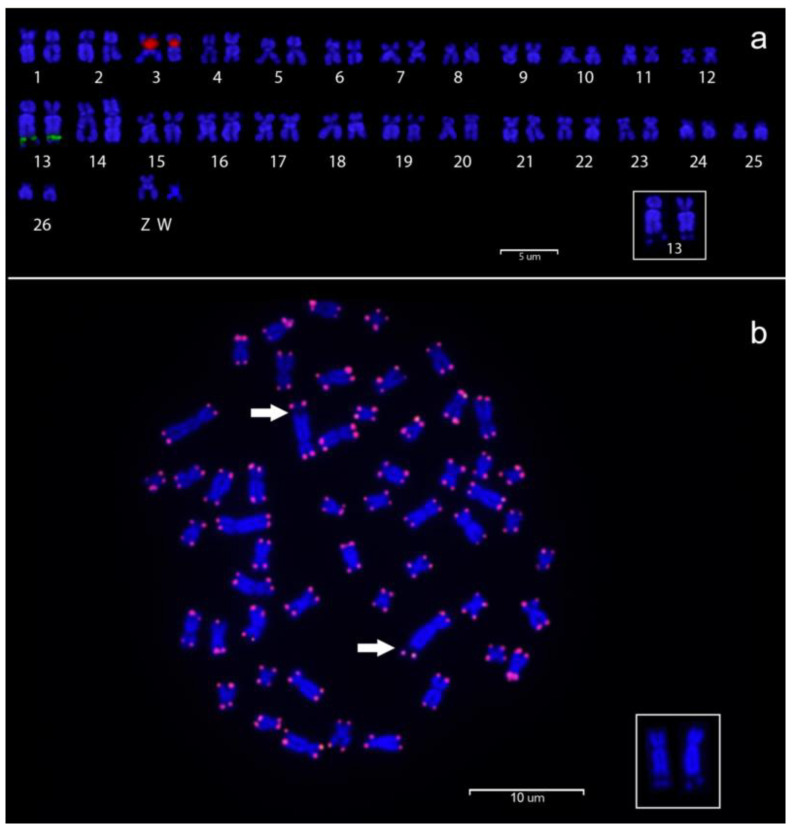
(**a**) *T. santarosensis* female karyotype following double FISH with 5S (red) and 18S (green) rDNA probes. (**b**) Metaphase plate after FISH using telomeric probes (TTAGGG)n represented by red signals in the terminal positions of all chromosomes. The arrows point to the secondary constriction associated with NORs. The insets show chromosome pair 13 of the same metaphases after DAPI.

**Figure 4 genes-14-01662-f004:**
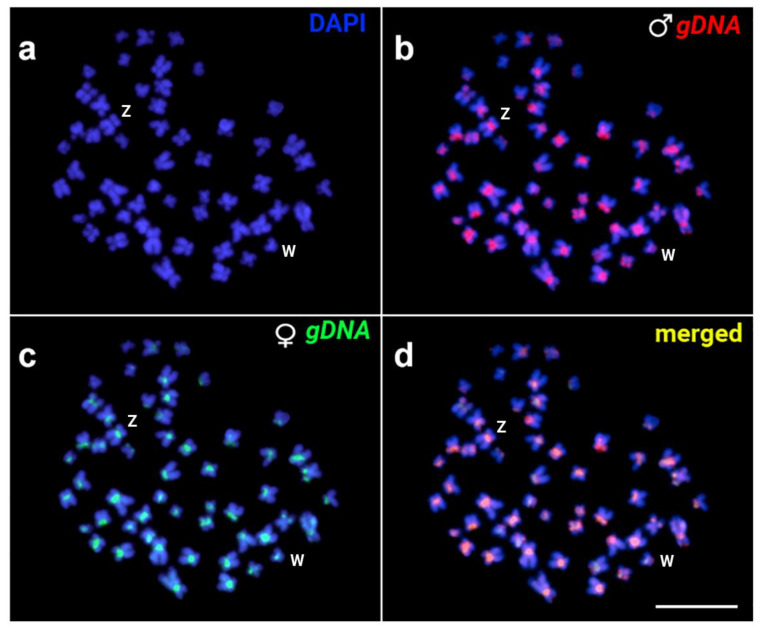
Female *T. santarosensis* metaphase plate after (**a**) DAPI and double CGH using (**b**) male (red signals) and (**c**) female (green signals) genomic DNA. In (**d**) merged signals. Scale bar = 10 µm. Z and W sex chromosomes are indicated.

## Data Availability

Sequences are deposited in GenBank (A.N. OR237841-43).

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
