# Peer review of "Integrating Genomic and Chromosomal Data: A Cytogenetic Study of Transancistrus santarosensis (Loricariidae: Hypostominae) with Characterization of a ZZ/ZW Sex Chromosome System"

_genes, 2023, doi:10.3390/genes14091662_

Round 1

Reviewer 1 Report

Many biodiversity hotspots have been subjected to years of habitat loss and environmental stresses, and many species are rapidly becoming extinct before they are recognized by humans. Freshwater fish species, in particular, are rapidly declining or even disappearing due to wading projects, excessive water use and pollution. Therefore, this study is of great significance for the study of sex determination mechanism of Transancistrus santarosensis (Loricariidae: Hypostominae), which enables researchers to have a deeper understanding of conservation genetics of this rare species with highly differentiated distribution.

The results of karyotype analysis in this study provide us with some valuable information. However, based on simple karyotyping photographs, it is not possible to confirm whether the sex chromosomes considered in the karyotyping formula given by the authors are sex chromosomes, or whether this difference will cause sex differences. Personally, more data needs to be added to make the results statistically significant. Alternatively, biological experiments should be added to provide direct evidence. Moreover, many of the claims in the Discussion Section lack the necessary results or data from other researchers to support them, and are so open that they must be narrowed down to what direct results can support.

Given that the samples involved in this study are rare species, it may be difficult to expand the sample size. Thus, it is suggested that the authors compress the manuscript from a full-length article into a communication format. Highlight the main results and possible assumptions, rather than based on such results, and discuss very academic issues more broadly in the discussion rather than focusing on the findings of this study.

In addition, there is a small problem LINE 193-195 if the results cannot or are not sufficient to be presented in the supplementary material, they should not be listed here, which is easy to mislead the readers.

English is generally good, minor revisions are recommended to improve readability.

Author Response

Reviewer 1

Many biodiversity hotspots have been subjected to years of habitat loss and environmental stresses, and many species are rapidly becoming extinct before they are recognized by humans. Freshwater fish species, in particular, are rapidly declining or even disappearing due to wading projects, excessive water use and pollution. Therefore, this study is of great significance for the study of sex determination mechanism of Transancistrus santarosensis (Loricariidae: Hypostominae), which enables researchers to have a deeper understanding of the conservation genetics of this rare species with the highly differentiated distribution.

The results of karyotype analysis in this study provide us with some valuable information. However, based on simple karyotyping photographs, it is not possible to confirm whether the sex chromosomes considered in the karyotyping formula given by the authors are sex chromosomes, or whether this difference will cause sex differences. Personally, more data needs to be added to make the results statistically significant. Alternatively, biological experiments should be added to provide direct evidence. Moreover, many of the claims in the Discussion Section lack the necessary results or data from other researchers to support them and are so open that they must be narrowed down to what direct results can support.

Given that the samples involved in this study are rare species, it may be difficult to expand the sample size. Thus, it is suggested that the authors compress the manuscript from a full-length article into a communication format. Highlight the main results and possible assumptions, rather than based on such results, and discuss very academic issues more broadly in the discussion rather than focusing on the findings of this study.

  • We appreciate the feedback and your recognition of the data's significance. We just disagree with the reviewer's point of view on sex chromosome confirmation. In fact, cytologically identifiable sex chromosomes are usually detected by karyotyping procedures. A sex-specific chromosomal polymorphism (in this case involving two chromosomes, one female-specific, W, and the other, Z, common to both sexes but with differential distribution among them, characterizes a ZW sex chromosome system. Our results allowed the identification of such heteromorphism between sexes both in Giemsa and C-banding karyotypes. There are many pieces of evidence in fishes to support the assumptions here presented, i.e. several cases of fish showing sex chromosomes identified by Giemsa and/or C-banding but without any sign of molecular differentiation detectable by CGH. See as examples
  • Marajó et al 2023 Journal of Fish Biology doi: 10.1111/jfb.15275
  • Sember et al 2021 Philosophical Transactions of the Royal Society B, https://doi.org/10.1098/rstb.2020.0098
  • Sassi et al 2020 Genes 2020 doi:10.3390/genes11101179
  • Xu et al 2019 International Journal of Molecular Sciences doi:10.3390/ijms20143571

We are conscious that more research is needed, to determine whether the sex-determination genes are similarly confined to one of these chromosomes. However, we believe that this is a separate concern:the identification of heteromorphic sex chromosomes is something different from  uncovering biological processes of sex determination. In the conclusion section (lines 386-389) we already reported “Further studies …. to investigate the presence of TEs and other repetitive sequences in the sex chromosomes here identified”.

We re-organized part of the discussion section providing supporting references when required 

In addition, there is a small problem LINE 193-195 if the results cannot or are not sufficient to be presented in the supplementary material, they should not be listed here, which is easy to mislead the readers.

We have deleted the sentence, as suggested. See present lines 204-206

Reviewer 2 Report

In the manuscript “Integrating Genomic and Chromosomal Perspectives: a cytogenetic study of Transancistrus santarosensis (Loricariidae: Hypostominae) with Characterization of a ZZ/ZW Sex Chromosome System” authors applied classical cytogenetic approaches to the plecos, fishes of the Loricariidae family. After a substantial introduction, the authors present the first cytogenetic analysis of T. santarosensis. The cytogenetic data is supplemented by the mitochondrial DNA sequence analysis. In addition to the accurate description of the male and female karyotypes, comparative genomic hybridization was used to identify sex chromosomes.

Authors represent high quality original data that is clearly summarized in 4 main and 1 supplementary figures and 2 supplementary tables. The manuscript is logically written and the data are convincing. However the following questions should be addressed to confirm the conclusions drawn:

Questions:

Q1. Please explain why individuals without an identified sex were included in the study if the main goal was to identify sex chromosomes.

Q2. Please clarify in the Results section why “15 µg of unlabeled blocking DNA (C0t-1) which was obtained by DOP-PCR from male genome” but not from female genome was used as a blocking DNA in CGH experiment (Figure 4). It would be also beneficial to include the result of the CGH experiment on male T. santarosensis metaphase plates in addition to CGH on female metaphase plates.

Q3. Lines 312-314: “On the other hand, the absence of molecular divergence between sex chromosomes, i.e., absence of a sex-specific region or the differential accumulation of repetitive DNAs, as herein showed by CGH” – It would be important to clarify why than the identified W chromosome in the female karyotype appears to be brightly stained by C-banding (Figure 2a).

Minor comments:

C1. My suggestion is to move Figure 1 to the supplementary material.

C2. I also suggest moving Figure S1 to the main text of the article.

C3. Figure 2. “Female (a) and male (b) T. santarosensis karyotypes arranged after sequential Giemsa (above) and C-banding (below) staining.” Instead of “above” and “below” consider labeling panels directly on the figure.

C4. Please reword phrases so that they do not contain expressions in brackets such as «whether (and which) sex», «sex chromosomes (if any) accumulated».

C5. Please indicate Z and W chromosomes on the Figure 4 for the CGH experiment.

Minor editing of English language is required.

Author Response

Reviewer 2

In the manuscript “Integrating Genomic and Chromosomal Perspectives: a cytogenetic study of Transancistrus santarosensis (Loricariidae: Hypostominae) with Characterization of a ZZ/ZW Sex Chromosome System” authors applied classical cytogenetic approaches to the plecos, fishes of the Loricariidae family. After a substantial introduction, the authors present the first cytogenetic analysis of T. santarosensis. The cytogenetic data is supplemented by the mitochondrial DNA sequence analysis. In addition to the accurate description of the male and female karyotypes, comparative genomic hybridization was used to identify sex chromosomes.

Authors represent high-quality original data that is clearly summarized in 4 main and 1 supplementary figures and 2 supplementary tables. The manuscript is logically written and the data are convincing. However, the following questions should be addressed to confirm the conclusions drawn:

Questions:

Q1. Please explain why individuals without an identified sex were included in the study if the main goal was to identify sex chromosomes.

  • The identification of sex chromosomes was one of the aims of our study, but as we wrote in the introduction (lines 107-109)“The primary aim of this work is to describe the unknown karyotype of the species and identify species-specific cytogenetic markers that will allow us to infer which are the plesiomorphic and derived characteristics within Hypostominae”. Thus, the maximum number of samples was used in order to confirm the general 2n and fish patterns and have more robust data.

Q2. Please clarify in the Results section why “15 µg of unlabeled blocking DNA (C0t-1) which was obtained by DOP-PCR from male genome” but not from female genome was used as a blocking DNA in CGH experiment (Figure 4). It would be also beneficial to include the result of the CGH experiment on male T. santarosensis metaphase plates in addition to CGH on female metaphase plates.

  • Some of us have large experience in performing CGH experiments in many different fishes (see among others de Moraes et al Frontiers in Genetics 2021, doi:13389/fgene.2021.769984; Yano et al Chromosome Res 2021, doi:10.1007/s10577-021-09674-1; Deon et al Genes 2020 10.3390/genes11111366, Barby et al Scientific Report 2019 doi:10.1038/s41598-019-38617-4) and also in other animal groups (like insects see Vidal et al Insects 2023, doi: 10.3390/insects14050440; or lizards see Spangenberg et al  Chromosoma 2020, doi:10.1007/s00412-020-00744-7 and Scientific Report 2020,  doi:10.1038/s41598-020-65686-7 ) and the chosen ratio of probe X Cot-ratio was largely tested in order to achieve the best results. In this specific case, we are searching for female-specific sequences, and thus we must use only MALE-specific blocking (to allow some putative female-specific ones to hybridize). We have added this sentence in the MM section:  “Such ratio of probe vs C0t-1 DNA amount was chosen based on experiments performed in other fish groups”. See present lines 180-181. Additionally, when ZW sex chromosomes are involved, it is useless to perform CGH experiments in male metaphase plates since no male-specific sequences are expected (as observed in the present case).

Q3. Lines 312-314: “On the other hand, the absence of molecular divergence between sex chromosomes, i.e., absence of a sex-specific region or the differential accumulation of repetitive DNAs, as herein showed by CGH” – It would be important to clarify why then the identified W chromosome in the female karyotype appears to be brightly stained by C-banding (Figure 2a).

  • We appreciate the comment. The presence of a prominent C-band on the W chromosome (indicating a highly heterochromatic content) does not imply that the sequences that comprise this heterochromatin are female-specific ones, as they can be also present in the male genome (probably in other chromosomal regions as well). This is not uncommon in fishes, because many species, despite having morphologically heteromorphic sex chromosomes, are at an early stage of their molecular differentiation. Se also the answer to Reviewer #1 and references reported therein. We have added this to the discussion section with a few references (present lines 337-351). If necessary we can add the above-cited references to the MS

Minor comments:

C1. My suggestion is to move Figure 1 to the supplementary material.

C2. I also suggest moving Figure S1 to the main text of the article.

  • These modifications were performed as suggested (Fig 1 to S1 and vice versa). See also lines 396-397

C3. Figure 2. “Female (a) and male (b) T. santarosensis karyotypes arranged after sequential Giemsa (above) and C-banding (below) staining.” Instead of “above” and “below” consider labeling panels directly on the figure.

  • These modifications were performed in Figure 2 as suggested.

C4. Please reword phrases so that they do not contain expressions in brackets such as «whether (and which) sex», «sex chromosomes (if any) accumulated».

  • We have adjusted these sentences as requested.

C5. Please indicate Z and W chromosomes in Figure 4 for the CGH experiment.

  • We have indicated them in the figure as suggested

 In addition to changes made in response to reviewers comments, we made also a few minor additional changes and corrected typos.

Round 2

Reviewer 1 Report

I have no principled opposition, only that the evidence is insufficient. I rarely see a regular original research paper that cites multiple references in its Conclusion Section. The Conclusion Section of the new version submitted by the author still cites a lot of references, because it is not enough to rely solely on the results of this paper to draw the conclusion given by the author. Thus, I still suggest that the editorial department asked the author revised the manuscript to give priority to in order to present objective results as a communication, rather than the full length article.

Minor editing of English language required

Author Response

Dear Reviewer 1,

We greatly appreciate your insightful and comprehensive review of our manuscript titled "genes-2526164", and we are committed to addressing your observations with a thoughtful and respectful response.

In consideration of your feedback, we would like to address the specific points you have raised about the conclusions:

  1. Conclusion About Cytogenetic Characteristics:

We re-organized the text of the conclusion section removing all the references. This section highlights both ancestral (chromosome number) and derived (heterochromatic blocks and localization of non-syntenic ribosomal genes) cytogenetic characteristics. The assertion that the karyotype of T. santarosensis remains unaffected by recent centric fusions and pericentromeric inversions, crucial mechanisms of chromosomal rearrangement, is well-supported by the absence of interstitial telomeric repeats and the consistency of diploid numbers with the ancestral group.

  1. Identification of ZZ/ZW Sex Chromosome System:

Our statement regarding the identification of an early-stage ZZ/ZW sex chromosome system is based on the presence of chromosomal heteromorphism, not only in size but also in the distribution of constitutive heterochromatin, and in the first response to the reviewer we provided references of similar evidences in other fishes.

  1. **Future Research Directions**:

The molecular identification of specimens confirmed accurate morphological attribution. However, this study solely utilized mitochondrial sequences of a single gene, as an extensive phylogenetic analysis was not the primary focus. A cytogenetic analysis of additional species within the Chaetostoma clade will provide valuable insights into whether the observed cytogenetic features in T. santarosensis are shared by other representatives of the genus and this specific clade. We considered the potential directions for future investigations, both including other. Hypostominae species from the Pacific slope and deepening the investigation of transposable elements (TEs) and other repetitive sequences within the identified sex chromosomes.

Regarding your suggestion of reorganizing the manuscript into a short note, we acknowledge your perspective. However, we respectfully express our differing opinion as we believe that our objective results are robust.

We are committed to further enhancing the manuscript based on your constructive input and eagerly anticipate your continued guidance.

Thank you once again for your time and expertise.

Mauro Nirchio